# HYPOGENEAGENT: HYPOTHESIS LANGUAGE AGENT FOR GENE-SET CLUSTER RESOLUTION SELECTION USING PERTURB-SEQ DATASETS

## ABSTRACT

Large-scale single-cell and Perturb-seq investigations routinely involve clustering cells and subsequently annotating each cluster with Gene-Ontology (GO) terms to elucidate the underlying biological programs. However, both stages, resolution selection and functional annotation, are inherently subjective, relying on heuristics and expert curation. We present **HYPOGENEAGENT**, a large language model (LLM)-driven framework, transforming cluster annotation into a quantitatively optimizable task. Initially, an LLM functioning as a gene-set analyst analyzes the content of each gene program or perturbation module and generates a ranked list of GO-based hypotheses, accompanied by calibrated confidence scores. Subsequently, we embed every predicted description with a sentence-embedding model, compute pair-wise cosine similarities, and let the agent referee panel score (i) the internal consistency of the predictions, high average similarity within the same cluster, termed intra-cluster agreement (ii) their external distinctiveness, low similarity between clusters, termed inter-cluster separation. These two quantities are combined to produce an agent-derived resolution score, which is maximized when clusters exhibit simultaneous coherence and mutual exclusivity. When applied to a public K562 CRISPRi Perturb-seq dataset as a preliminary test, our Resolution Score selects clustering granularities that exhibit alignment with known pathway compared to classical metrics such silhouette score, modularity score for gene functional enrichment summary. These findings establish LLM agents as objective adjudicators of cluster resolution and functional annotation, thereby paving the way for fully automated, context-aware interpretation pipelines in single-cell multi-omics studies.

## 1 INTRODUCTION

High-throughput single-cell technologies now profile hundreds of thousands of cells in a single experiment, revealing cellular heterogeneity at unprecedented resolution. A cornerstone of every single-cell pipeline is clustering, the partitioning of cells into transcriptionally coherent groups that serve as proxies for cell types, states or genetic perturbations. The resolution parameter of graph-based community detection (e.g. Leiden or Louvain Traag et al. (2019)) directly controls how many clusters are returned: low values produce a few coarse partitions, whereas high values yield a fine-grained mosaic. Choosing the right resolution is therefore critical, as it determines not only the granularity of biological discovery but also the downstream functional annotation of each cluster.

Classical metrics such as modularity, silhouette score and cluster stability offer generic notions of statistical quality, yet they ignore the fact of whether the resulting clusters are biologically interpretable. In practice, investigators inspect marker-gene heatmaps, adjust the resolution until clusters look clean, and then assign Gene-Ontology (GO) terms manually. This procedure is inherently subjective, prone to human bias, and is poorly reproducible across annotators, laboratories and datasets. Recent studies have shown that Large Language Models (LLMs) can reason over gene sets and generate plausible GO annotations Hu et al. (2025); Wang et al. (2025); Wu et al. (2025), but a principled way to use these models for resolution selection has not been explored. Consequently, the field still lacks a quantitative, biology-aware criterion that bridges unsupervised clustering with

automated annotation. Beyond single-cell biology, there is a growing wave of research exploring AI agents for scientific discovery, such as AlphaEvolve for algorithmic innovation Novikov et al. (2025), ROBIN for multi-agent collaboration in automating scientific discovery Ghareeb et al. (2025), and frameworks envisioning the AI co-scientist paradigm Gottweis et al. (2025). Similar domain-specific agents have also emerged in biomedicine, including BioDiscoveryAgent for perturbation experiment design Roohani et al. (2024) and primers discussing the role of language models in biological research Simon et al. (2024). These works highlight an ongoing shift from static predictive models to interactive agents capable of hypothesis generation and experimental guidance.

Specifically, the silhouette score Lovmar et al. (2005) is one of the most widely used, model agnostic diagnostics for assessing the quality of a partition produced by an arbitrary clustering algorithm. Unlike indices that rely on graph-theoretic quantities (e.g. modularity) or external ground truth, the silhouette exploits only pair-wise distances in the original feature space and can therefore be applied to any embedding (full expression matrix, PCA, UMAP, neighborhood graph). Clustering methods that operate on a graph of k-nearest neighbors(kNN)(e.g.Louvain or Leiden) are often evaluated with the modularity index Newman (2006). Modularity quantifies how much the density of edges within clusters exceeds the density expected in a random graph with the same node–degree distribution, thus rewarding partitions that form well-connected communities on the graph. Functional enrichment analysis use the Fisher-exact framework to give a ranked list of GO terms with Benjamini–Hochberg–adjusted $p$-values ($P_{\mathrm{adj}}$), terms with $P_{\mathrm{adj}} < 0.05$ were considered significant.

Here we introduce **HYPOGENEAGENT**, an agent-based annotation consistency framework that closes this gap. Inspired by the broader movement of AI agents for science Novikov et al. (2025); Ghareeb et al. (2025); Gottweis et al. (2025); Roohani et al. (2024); Simon et al. (2024),we treat an LLM augmented with domain databases and self-verification prompts as a gene-set analyst Hu et al. (2025); Wang et al. (2025) tasked with describing the dominant biological process for every cluster. From the resulting GO hypotheses and confidence scores, we derive two complementary metrics: Intra-cluster agreement, the degree to which all cells inside a cluster support the same GO explanation, and Inter-cluster distinctiveness, the extent to which different clusters receive different explanations. Combining the two yields a Resolution Score that is maximized when clusters are simultaneously coherent and mutually exclusive in their biological function. The score is computed automatically for a grid of resolution values, allowing an objective choice of clustering granularity without manual marker-gene inspection.

We validate our method using public Perturb-seq datasets of K562 cells targeting disease pathways Replogle et al. (2022). Across all benchmarks, the agent-derived Resolution Score selects parameter settings that recover known perturbation effects better than modularity and silhouette criteria, while remaining computationally efficient. The approach is model-agnostic and readily extends to multi-omics modalities by feeding modality-specific summaries to the same gene agent.

In summary, we propose a **HYPOGENEAGENT** that couples single-cell clustering with automated functional annotation; provide formal definitions of intra-cluster agreement and inter-cluster distinctiveness; define a single Resolution Score that turns subjective resolution tuning into a data-driven optimization; and apply comprehensive validation on large perturbation datasets, demonstrating superior biological interpretability compared with traditional metrics. This work establishes a general methodology for integrating LLM reasoning with quantitative genomics, paving the way for fully automated, biology-aware single-cell analytics.

## 2 RELATED WORK

**Resolution selection in single-cell clustering** Cell Ranger Zheng et al. (2017), the analysis pipeline for processing droplet-based scRNA-seq data, after PCA and $k$-NN graph construction it initially set default Leiden/Louvain-resolution values, but users quickly realized that biologically meaningful partitions require data-specific tuning. Generic statistical indices such as silhouette width, Calinski-Harabasz and Davies–Bouldin are widely used in Seurat Butler et al. (2018) and Scanpy Wolf et al. (2018). MultiK Liu et al. (2021) is a tool for objective selection of insightful Ks and achieves high robustness through a consensus clustering approach in scRNA-seq data. Although useful, these approaches remain agnostic to biological interpretation.

**Automated cluster annotation** Early tools such as scmap Kiselev et al. (2018) and CellAssign Zhang et al. (2019) map clusters onto a reference atlas via marker-gene enrichment. Functional annotation is typically performed post hoc with over representation analysis such as GSEA-P Subramanian et al. (2007), none of which feeds back into the resolution choice. Consequently, manual inspection of heat maps and dot plots remains common practice.

**Large language models in computational biology** LLMs are now being used at almost every layer of the single-cell/perturb-seq analysis stack. Beyond literature triage and protein-property prediction, Chen et al. Chen & Zou (2025) showed that a ChatGPT-distilled embedding (no hand-engineered features) can rival scVI in representing cell states, providing an LLM-native feature space for downstream tasks. Yuksekgonul et al. Yuksekgonul et al. (2025) demonstrated how reinforcement learning with LLM-based rewards markedly improves biological text generation, suggesting a route to further tune domain-specific agents. Several recent works introduce multi-step agents that delegate planning and retrieval to an LLM. BioDiscoveryAgent Roohani et al. (2024) formulates CRISPR-screen design as an agentic reasoning problem, while PerTurboAgent Hao et al. (2025) builds a self-planning loop that proposes follow-up Perturb-seq experiments. Gonzalez Gonzalez et al. (2025) couples causal neural networks with GPT-4 rationales to forecast synergistic drug–gene interventions, and Biomni Huang et al. (2025) compresses biological knowledge into an open source model with 1 b parameter, lowering the entry barrier for the development of internal agents. Together with Hu et al. Hu et al. (2025) evaluated GPT-4 and Claude on explaining gene sets, showing that chain-of-thought improves factual accuracy; GeneAgent Wang et al. (2025) introduced a self verification loop and database retrieval, achieving state-of-the-art GO term generation; Wu et al. Wu et al. (2025) demonstrated that instruction tuned LLMs can recover perturbation mechanisms directly from PubMed abstracts, these works highlight the momentum toward LLM-centric pipelines. Yet all current systems act after the clusters are fixed; none feeds functional feedback back into the clustering hyper parameters.

## 3 METHOD

### 3.1 OVERVIEW

**HYPOGENEAGENT** implements a multi-stage workflow. First, it generates candidate clusterings of Perturb-seq data across a grid of resolution parameters. Next, a large language model (LLM) analyzes each cluster's gene signature and proposes functional descriptions in terms of the underlying biological processes. These descriptions are then processed to extract embeddings and compute annotation-consistency metrics that identify the clustering resolution hyperparameter yielding clusters that are internally coherent yet externally distinct. In doing so, **HYPOGENEAGENT** closes the loop between unsupervised partitioning and biologically informed interpretation.

### 3.2 CLUSTERING PROCEDURE

After basic data processing we performed the following steps: Scaling and dimensionality reduction, Multi-resolution community detection, Gene-to-cluster assignment matrix (3000 × 10), and Perturbation-to-cluster assignment matrix (2005 × 10). Please refer to the Clustering procedure session in the appendix for more details.

### 3.3 AGENT-BASED ANNOTATION

After clustering, every cell cluster (or perturbation cluster) was represented by its gene-set signature, the most over-expressed genes ranked by log-fold-change against the remaining cells. We fed each signature to an autonomous HypoGeneAgent instance, implemented on the top performance model from stage one's benchmark, the GPT-o3 model with chain-of-thought and self-verification prompts. For completeness, we briefly recap the agent workflow:

**Evidence retrieval** For every gene in the signature the agent calls a retrieval tool that surfaces concise functional summaries from GO, KEGG and PubMed. Retrieved snippets are appended to the system prompt.

Table 1: Metrics for evaluating agent-generated GO annotations($k$ is cluster).

| Symbol | Purpose | Definition |
|---|---|---|
| $S_k^{\cos}$ | Semantic accuracy | Cosine similarity between the reference text $g_k$ and the agent text $h_k$. |
| $\text{ICS}_{h,k}$ | Intra-cluster agreement | Cosine distance between the top hypothesis $h_{k1}$ and each of the remaining hypotheses $h_{k2}, \ldots, h_{k5}$ produced for the same cluster. |
| $\text{ICD}_k$ | Inter-cluster distinctiveness | Mean pair-wise cosine distance between the top hypothesis of cluster $k$ and the top hypotheses of all other clusters at resolution $r$. |

**Hypothesis generation** For each cluster, the agent returns up to $H = 5$ candidates. Each candidate $h_{ki}$ is accompanied by (i) a description in plain English, similar to the Gene-Ontology Biological-Process (GOBP) brief description, and (ii) a calibrated confidence score $c_{ki} \in [0, 1]$.

**Output set** The complete output for cluster $k$ is $\mathcal{H}_k = \big\{(h_{k1}, c_{k1}), (h_{k2}, c_{k2}), \ldots, (h_{k5}, c_{k5})\big\}$, where the pairs are ordered by decreasing confidence ($c_{k1} \geq c_{k2} \geq \ldots$).

## 3.4 METRICS

We assess the agent-generated annotations on the three orthogonal axes listed in Table 1. We now detail the computation of the third axis, the semantic-similarity term $S_k^{\cos}$.

**Cosine similarity of annotations** We embed every free-text annotation with the OpenAI text-embedding-3-large (i) the agent's best hypothesis for cluster k (ii) the reference description (curated GO term or expert label) To be two vectors, the raw semantic score is the cosine similarity of these two vectors and lies in [0,1] where 1 indicates a highly significant match and 0 indicates no better than random.

**Intra-cluster similarity (ICS)** For a given clustering resolution $r$ the agent returns, for every cluster $k$, a ranked list of up to five GO hypotheses $h_{k1}, \ldots, h_{k5}$. To quantify how consistently these hypotheses describe the same biological topic we compare each of the four lower-ranked hypotheses with the top hypothesis $h_{k1}$. The average intra-cluster agreement is therefore $\text{ICS}_k = \frac{1}{4} \sum_{h=2}^{5} \text{sim}(h_{k1}, h_{kh})$, where $\text{sim}(\cdot, \cdot)$ is the cosine similarity between the sentence-embedding vectors of two hypothesis texts. A high $\text{ICS}_k$ indicates that all agent-generated explanations for cluster $k$ converge on the same biological theme, implying that the cluster is internally coherent and biologically robust.

**Inter-cluster distinctiveness (ICD)** For resolution $r$ we also ask how different a given cluster $k$ is from all other clusters. Let $h_{k1}$ be the top-ranked hypothesis for cluster $k$ and $h_{\ell1}$ the top hypotheses of every other cluster $\ell \neq k$. The mean pair-wise similarity is $\text{ICD}_k = \frac{1}{C-1} \sum_{\ell \neq k} \text{sim}(h_{k1}, h_{\ell1})$, where $C$ is the total number of clusters and $\text{sim}(\cdot, \cdot)$ denotes cosine similarity. A lower $\text{ICD}_k$ therefore implies that cluster $k$ is well separated from the rest in terms of biological interpretation.

**Resolution Score** To combine internal coherence and external separation we define, for each cluster $k$, $\text{RS}_k = w\overline{\text{ICS}}_k + (1 - w)(1 - \text{ICD}_k), 0 \leq w \leq 1$. Here $\overline{\text{ICS}}_k$ is the average intra-cluster agreement and $1 - \text{ICD}_k$ rewards distinctiveness (large when clusters are dissimilar). We adopt $w = \frac{1}{3}$, i.e. one-third weight on agreement and two-thirds on distinctiveness, which was chosen by a small grid search and found to give a stable ordering of resolutions across data sets. A higher $\text{RS}_k$ indicates a cluster that is (i) internally convergent and (ii) externally dispersed, making it a strong candidate for a biologically meaningful partition.

## 3.5 TRADITIONAL METHODS

We benchmarked our method with these traditional methods: silhouette score Lovmar et al. (2005), modularity score Newman (2006), and functional enrichment analysis. Please refer to the Traditional methods session in the appendix for more details.

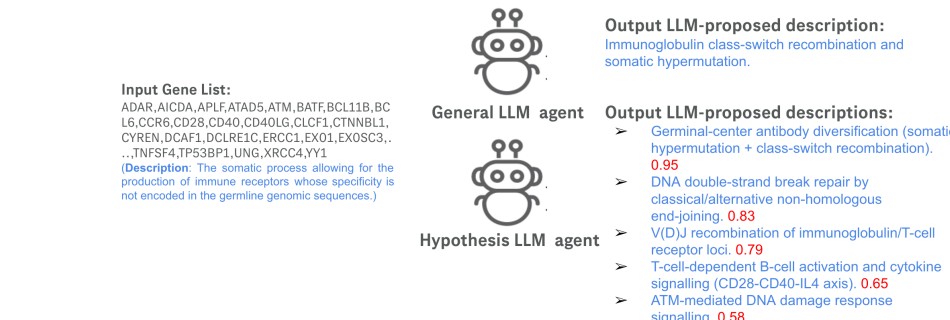

Figure 1: Illustration of the two prompt designs. **Top**: the *general-analysis* prompt encourages a free-text GO explanation. **Bottom**: the *hypothesis* prompt elicits up to five ranked GO explanations with confidence scores.

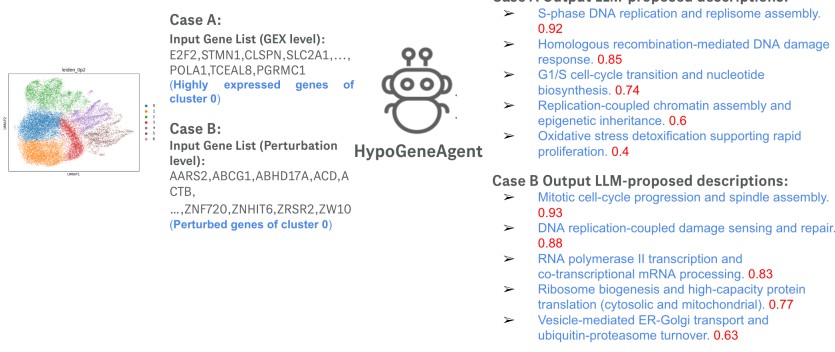

Figure 2: Illustration of the HypoGeneAgent.

## 4 EXPERIMENT

### 4.1 EXPERIMENTAL SETUP

We designed experiments to (i) assess the agent's ability to recover known processes on curated Gene Ontology Biological Processes (GOBP) sets and (ii) evaluate **HYPOGENEAGENT** for resolution selection on Perturb-seq. Stage 1 and Stage 2 here denote our research protocol—configuration selection and fixed-configuration deployment. Figure 1 contrasts a general-analysis prompt, which yields a single free-text explanation, with a hypothesis prompt that returns up to five ranked GO terms with calibrated confidence scores. Figure 2 summarizes the end-to-end workflow, from multi-resolution clustering and signature construction to agent annotation and resolution scoring.

### 4.2 IMPLEMENTATION

**Stage 1: Parameter benchmark on curated GOBP gene sets**

**Data** We used the 100 non-redundant GOBP gene sets as a clean, reference-labeled benchmark (Data & code session in the appendix).

**LLM Agent backbone** The default model is GPT-4o (2024-08-01-preview), accessed through Azure OpenAI. Alternative back-ends evaluated were GPT-o3, GPT-5, Gemini-2.0-flash and Gemini-2.5-pro.

**Prompt engineering** We implemented a prompt factory that produces two canonical prompt classes: (i) General-analysis prompt: used to test basic reasoning and retrieve free-text GO explanations. The agent only needs to propose one candidate prediction for each input gene set list. There are two versions prompt provided (V2 is improved with more details instructions than V1). (ii) Hypothesis prompt: encourages ranked, confidence-weighted GO hypotheses. The agent needs to propose top

5 candidate predictions for each input gene set list. All the prompts templates relevant to this work are in the appendix session.

**Text embedding method** The default embedding is OpenAI text-embedding-3-large, accessed through Azure OpenAI. Alternative methods evaluated were SapBERT Lim & Kim (2022) and Nomic AI Nussbaum et al. (2024).

**Temperature** Range [0,1] with step 0.1.

**Stage 2: Agent-guided resolution selection on Perturb-seq**

**Data** The K562 Perturb-seq dataset was processed with Scanpy 1.9.6 as a benchmark (Data & code session in the appendix).

**Application of the hypothesis prompt** Based on the conclusions of Stage 1 of Experiments, we selected a single prompt/model/embedding configuration and *held it fixed* for all downstream analyses. For every cluster in gene-expression space and every perturbation group in CRISPR-guide space, we constructed a cluster-specific gene signature (ranked by positive log fold-change) and submitted it to **HYPOGENEAGENT**. The agent returned up to five GO hypotheses with confidence scores for each cluster. Using these descriptions, we computed intra-cluster agreement (ICS), inter-cluster distinctiveness (ICD), and the resulting *Resolution Score* on the fly across the full resolution sweep (10 resolutions, up to 20 clusters).

### 4.3 ABLATION & MAIN RESULTS

**Stage 1: Parameter benchmark on curated GOBP gene sets**

**Compare Embedding Methods** We compared the performance of three different types of embedding methods by fixing the LLM as GPT-4o, with temperature as 0 and the V1 general prompt design. Figure S1a shows different embedding methods (OpenAI embedding, SapBERT, Nomic AI) can be regarded as different rulers, each one has its own best, worst and median similarity score range, so it is important to keep consistent usage of the specific embedding methods for reasonable benchmark work. We adopt OpenAI embedding as the default one for all other benchmarks.

**Compare general prompt method V1 & V2 (improved instruction)** Figure S1b shows the improved instruction of general prompt can help nonthinking LLMs (GPT-4o, Gemini-2.0-flash) get better performance. Besides, thinking LLMs (GPT-o3, Gemini-2.5-pro) show better and more stable performance on both V1 and V2 general prompt cases.

**Compare T parameter & repeat on GPT-4o model** Figure S1c shows temperature T influences very little on nonthinking LLM GPT-4o for general prompt case; the repeat test results are similar to each other. It shows the GPT-4o model has stable performance when adjusting T in range [0,1] and repeat 3 times for each T. It is worth testing how these will influence the performance of thinking LLMs in the next step.

**Compare top5 candidates on GPT-o3 model** Figure S1d shows the group performance of top 5 proposed candidates by GPT-o3 model with the hypothesis prompt method. The candidates were ranked by the confidence score of the model itself. The top 1 candidate group shows the highest median cosine similarity score with the ground truth compared with all other candidate groups, which is not only reasonable as our expectation, but also a validation of the model's ability to generate hypotheses and self-judgement.

**Compare general V2 & hypothesis prompt methods on different LLMs** Figure S1e shows that for both general prompt case and hypothesis prompt case, thinking LLMs perform better than non-thinking LLMs; GPT-5 performs good but not as stable as expected; thinking LLM GPT-o3 shows the best performance among these LLMs currently, especially good at the hypothesis task.

**Other metrics for measuring the performance of LLMs** In Figure S2, AUC metric for GPT-4o top1 group and GPT-o3 top1 group performance are compared at different thresholds. The higher the AUC, the better the performance or accuracy of the model's prediction. Model GPT-o3 shows better performance at all the thresholds, performs best (AUC = 0.743), especially when the threshold is 0.40, which is also consistent with the median of the cosine similarity score being between 0.4 to 0.5 (Figure S1 d). Figure S3 shows the consistency comparison between similarity score & model's confidence score. Interestingly, GPT-o3 has very consistent self confidence score judgement with

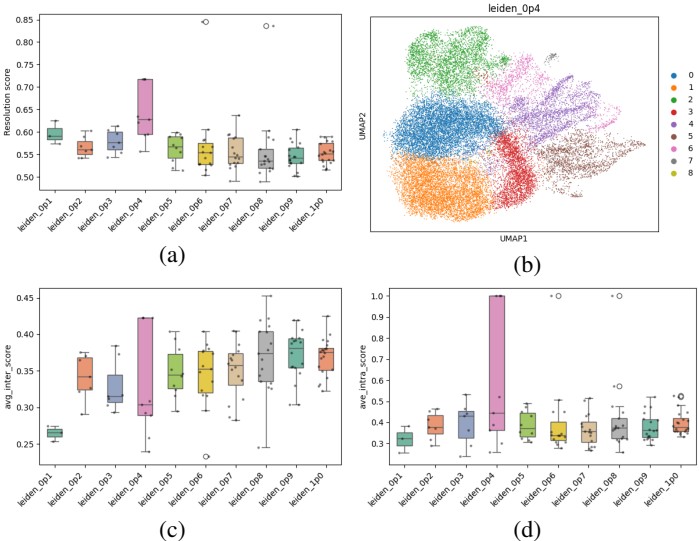

Figure 3: Application of HypoGeneAgent on cluster resolution selection – GEX level. **(a,c,d)** Box plot of the resolution score (a), the average inter score (c), and the average intra score (d) of each cluster for each leiden resolution case [0.1, 1.0]. **(b)** UMAP of clustering results at leiden resolution 0.4.

the semantic similarity score (ground truth) (Figure S3a); GPT-5 also shows a good consistency comparison between the similarity score and its confidence score.

In summary, a combination of thinking LLMs with a reasonable hypothesis prompt method is a good strategy to achieve ideal performance of the gene set annotation prediction task.

**Stage 2: Agent-guided resolution selection on Perturb-seq**

**Application on cluster resolution selection – GEX level**

For every gene-expression (GEX) cluster, we extracted its set of markers, the genes showing the highest positive log fold change and supplied this list to **HYPOGENEAGENT**. Using the hypothesis prompt, the agent returned up to five ranked GO annotations with calibrated confidence scores. From these outputs, we computed the intra-cluster agreement, the inter-cluster distinctiveness and the Resolution Score. Figure 3a visualizes, for each Leiden resolution parameter $r \in [0.1, 1.0]$, the distribution of $\mathrm{RS}_k$ across all clusters. Because a higher score indicates a partition that is both internally coherent and externally distinct, the optimal resolution is the one with the highest median score; **HYPOGENEAGENT** selects $r = 0.4$. Figure 3b displays the UMAP embedding colored by the Leiden-0.4 labels, revealing nine well-separated clusters in agreement with the score-based choice. To confirm the contribution of the two components individually, Figure 3c plots the average inter-cluster distinctiveness and Figure 3d the average intra-cluster agreement for the same grid of resolutions. The best solution is characterized by a low inter-cluster score (distinct clusters) and a high intra-cluster score (coherent clusters), both of which again peak at $r = 0.4$. Thus, the independent metrics and their combined Resolution Score converge on the same, biologically meaningful clustering granularity.

**Application on cluster resolution selection – perturbation level**

For each perturbation level cluster, we can extract the perturbed gene labels list as the input of the **HYPOGENEAGENT**. Then the **HYPOGENEAGENT** can propose top 5 annotation candidates according to the instruction of the hypothesis prompt to illustrate the biology function of this cluster-specific perturbed genes module. Next, the relevant metrics and scores will be computed for each cluster at each resolution. Figure 4a shows the box plot of the resolution score of each cluster for each leiden resolution parameter case $r \in [0.1, 1.0]$(with the $w = \frac{1}{3}$), the best leiden resolution chosen by **HYPOGENEAGENT** is 0.5 based on the resolution score. Figure 4b shows the UMAP of clustering results at leiden resolution 0.5, there are 10 clusters at this resolution, and it is as clear as expected. All of the other UMAPs at other resolutions are shown in Figure S4. Figure 4c and Figure 4d show the box plot of the average inter score of each cluster for each leiden resolution parameter

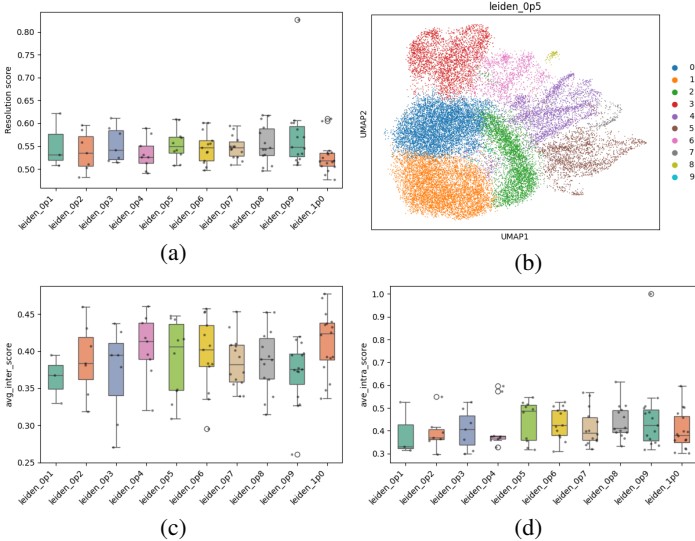

(a)

(b)

(c)

(d)

Figure 4: Application of HypoGeneAgent on cluster resolution selection – perturbation level. **(a,c,d)** Box plot of the resolution score (a), the average inter score (c), and the average intra score (d) of each cluster for each leiden resolution case [0.1, 1.0]. **(b)** UMAP of clustering results at leiden resolution 0.5.

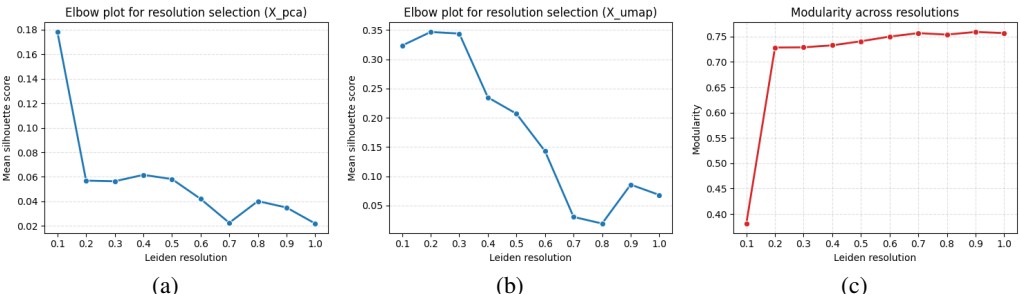

(a)

(b)

(c)

Figure 5: Traditional methods for the clustering judgment.. **(a)** Elbow plot of Silhouette score for each Leiden resolution ($X_{\mathrm{pca}} = 40$). **(b)** Elbow plot of Silhouette score for each Leiden resolution ($X_{\mathrm{umap}}$). **(c)** Elbow plot of modularity score for each Leiden resolution.

case and the box plot of the average intra score of each cluster for each leiden resolution parameter case $r \in [0.1, 1.0]$ separately, similarly, the best selected leiden resolution is still 0.5, consistent with the combination of both items of resolution score. But how does the hyper parameter w influence the resolution score ? We did the hyper parameter w tests in range [0,1] (Figure S5). It shows for different clusters, the tendency of resolution score changing with w can be different, those outliers can be the key clusters to be explored further in biology level.

## 4.4 COMPARATION WITH TRADITIONAL METHODS

### 4.4.1 SILHOUETTE SCORE

This section clarifies why the silhouette remains a popular baseline and sets the stage for our biologically informed alternative. Figure 5 a and b show the resulting elbow curves, which elbow at resolutions 0.5 and 0.6, respectively. Although the silhouette peak or elbow can provide a useful sanity-check, they do not incorporate biological knowledge. The silhouette assumes convex cluster geometry; elongated or manifold-shaped clusters can lead to deceptively low scores. Moreover, $S$ is sensitive to the choice of distance metric and the presence of noise features. These weaknesses motivate the integration of domain-specific annotation consistency, as implemented in **HYPOGENEAGENT**.

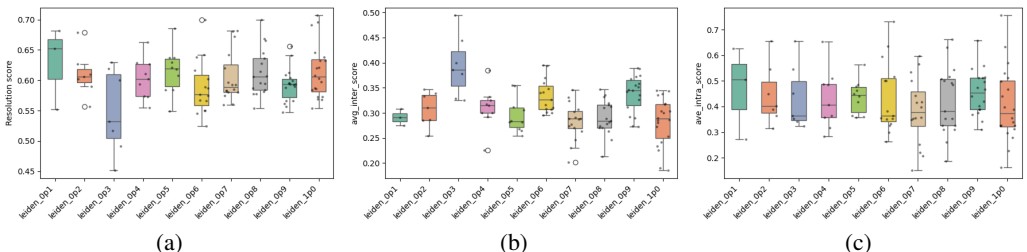

Figure 6: Gene Ontology (GO) enrichment analysis. **(a)** Box plot of the resolution score. **(b)** Box plot of the average inter score. **(c)** Box plot of the average intra score.

### 4.4.2 MODULARITY SCORE

The modularity curve $Q(r)$ is shown in Fig. 5c. A clear maximum is observed at $r = 0.7$, but the increase beyond $r = 0.5$ is marginal. Modularity is insensitive to clusters smaller than $\sqrt{2m}$ nodes and may favor mergers of biologically distinct micro-clusters. Moreover, it ignores gene-expression coherence; a partition with high $Q$ can still mix unrelated cell states if those states happen to be densely connected in the kNN graph. These shortcomings motivate the biology-aware Resolution Score we introduced, which combines intra-cluster agreement of functional annotations with inter-cluster distinctiveness.

### 4.4.3 FUNCTIONAL ENRICHMENT ANALYSIS

Fig. S6a summarizes the top enriched functions of cluster 0 at resolution 0.1, Fig. S6b summarizes the top enriched functions of cluster 0 at resolution 0.4. By applying the similar metrics raised for **HYPOGENEAGENT** on these enrichment results, we got the box plot of the resolution score in Fig. 6a, the box plot of the average inter score in Fig. 6b, and the box plot of the average intra score in Fig. 6c, consider the reasonability of cluster numbers we expected, so the selected resolution can be 0.5 or 0.4, which is consistent with our previous selection with **HYPOGENEAGENT**. Taken together, the enrichment analysis validates that the clusters produced at the Resolution Score maximum are biologically coherent and align with the expected results from **HYPOGENEAGENT**, underscoring the utility of **HYPOGENEAGENT** for simultaneous resolution selection and cluster interpretation.

In summary, deploying **HYPOGENEAGENT** on K562 Perturb-seq data produced an objective Resolution score curve whose optimum matched known perturbation biology and exceeded traditional metrics such as modularity, silhouette score and functional enrichment analysis. The same pipeline simultaneously generated unbiased GO annotations for every cluster in minutes, orders of magnitude faster than manual curation.

## 5 CONCLUSION

In this work, we proposed **HYPOGENEAGENT**, systematically dissected the design space of LLM-based gene-set interpretation and demonstrated how an agent architecture coupled with an optimized hypothesis prompt transforms both cluster annotation and resolution selection in single-cell/perturb-seq studies. In summary, there are several advantages of **HYPOGENEAGENT**: (i) Up-to-date biological knowledge. (ii) Reduced human bias and higher throughput. (iii) Seamless resolution selection. The same agent output drives the intra-/inter-cluster consistency metrics, turning an otherwise heuristic parameter search into a quantitative, biologically informed optimization. Together, these results position **HYPOGENEAGENT** as a powerful, general-purpose tool for single-cell, perturb-seq and multi-omics analyses, capable of both objective resolution tuning and rapid, bias-free automated functional annotation by leveraging LLMs. However, several limitations remain: Larger atlases such as the Human Cell Atlas (millions of cells) or whole-genome CRISPR screens will be necessary to test scalability and statistical power; Generalizability to other ontologies and modalities; LLM dependence and cost; prompt sensitivity are also necessary to be considered in the future.

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

## A  APPENDIX

## B  DATA & CODE

**Gene-Ontology Biological-Process (GOBP) reference sets** We obtained the complete set of Biological-Process (BP) terms from the Gene Ontology Consortium (release 2024-03-01). For every BP term we built a "gene set" consisting of all genes annotated to that term "or any of its child terms" (true-path rule).

**K562 Perturb-seq dataset  Replogle et al. (2022)** Raw count matrices and metadata for the CRISPR-i Perturb-seq screen of essential genes in K562 cells were downloaded from Zenodo. The experiment targets  5 000 genes with three guide RNAs each and includes non-targeting controls (NTCs). We processed the data as follows: (i) Subset the dataset with specific batches (including 1,6,8,9,10,20,30,36,40) (ii) Quality control: drop cells with $<$200 genes, drop genes expressed in $<$3 cells, and keep cells with  10% mitochondrial counts, yielding 25,161 high-quality cells. (iii) Expression normalization: library-size normalization to 10 000 counts per cell and log1p transformation; 3000 highly-variable genes were selected. The resulting AnnData object (25161 cells $\times$ 3000 genes) is used for all clustering and agent-based annotation experiments presented in this study. Both the processed AnnData file and the GOBP reference gene-set library are available at our GitHub repository to ensure full reproducibility.

We will release code upon acceptance (subject to institutional approval).

## C  CLUSTERING PROCEDURE

**Scaling and dimensionality reduction** Expression values were z-scored gene-wise, a 40-component PCA was fitted, and a 10-nearest-neighbor graph was built in the PCA space (a default parameters setting for a standard benchmark).

**Multi-resolution community detection** We ran the Leiden algorithm at ten granularities (resolution parameter r in 0.1,0.2,...,1.0). Each run wrote its cluster labels, e.g. leiden_0p4 for (r=0.4). Across this grid the number of clusters ranged from 3 (r=0.1) to 20 (r=1.0), with the smallest cluster size reported for every run to ensure adequate cell counts.

**Gene-to-cluster assignment matrix (3000 $\times$ 10)** For every resolution, we calculated the mean expression of each highly-variable gene across all clusters and assigned the gene to the cluster with the highest mean. Concatenating the ten resolution-specific assignments produced a 3000 genes $\times$ 10 resolutions categorical matrix that indicates where each gene is maximally expressed.

**Perturbation-to-cluster assignment matrix (2005 $\times$ 10)** Using the same cluster labels, we grouped cells by CRISPR guide, and for every resolution, recorded the modal (most frequent) cluster identity within each perturbation. This yielded a 2005 perturbations $\times$ 10 resolutions table that links each perturbation to its dominant transcriptional neighbourhood. These two matrices constitute the input for the agent-based annotation consistency scoring.

## D  TRADITIONAL METHODS

**Silhouette score :  a geometry based indicator of cluster quality** For every sample $i$ let $a(i)$ be the mean distance from $i$ to all other points in the same cluster and $b(i) = \min_{C \neq c_i}\{$ mean distance from $i$ to cluster $C\}$ be the "nearest-neighbor" distance to the next-best cluster. The individual silhouette value is

$$s(i) = \frac{b(i) - a(i)}{\max\{a(i),\, b(i)\}} \in [-1, 1].$$

A value close to 1 indicates that the point is well matched to its own cluster and poorly matched to neighboring clusters; a value near 0 implies that the point lies on the decision boundary; negative values suggest a possible misassignment. The global silhouette index of a partition is the arithmetic mean $S = \frac{1}{n}\sum_{i=1}^{n} s(i)$.

For each resolution $r$ we evaluate the silhouette on the 40-dimensional PCA representation ($X_{\text{pca}}$) as well as on the two-dimensional UMAP embedding ($X_{\text{umap}}$). The per-cell scores are obtained with the Scanpy function sc.metrics.silhouette(adata, groupby="leiden_r", obsm="X_pca"); the overall score $S_r$ is their mean.

Advantages: 1. Scale-free interpretability. Because $s(i)$ is normalised to $[-1, 1]$, scores can be compared across datasets and distance metrics without additional calibration. 2. Sensitivity to both cohesion and separation. Many indices capture only one aspect; the silhouette simultaneously penalizes low intra-cluster density ($a(i)$) and low inter-cluster separation ($b(i)$). 3. No distributional assumptions. Applicable to Euclidean, cosine or even pre-computed graph distances, making it attractive for high-dimensional single-cell embeddings. 4. Elbow diagnostics. Plotting $S$ over a range of hyper-parameters (e.g. Leiden resolution $\gamma$) often exhibits an elbow or a peak which can guide the choice of granularity.

**Modularity score : a graph-based measure of community structure** Let $G = (V, E)$ be an undirected weighted graph with adjacency matrix $A_{ij}$, node degree $k_i = \sum_j A_{ij}$ and total edge weight $m = \frac{1}{2} \sum_{ij} A_{ij}$. For a partition $\{c_1, \ldots, c_C\}$ the Newman–Girvan modularity is

$$
Q = \frac{1}{2m} \sum_{i,j} \Big( A_{ij} - \gamma \frac{k_i k_j}{2m} \Big) \delta(c_i, c_j), \qquad \delta(c_i, c_j) = \begin{cases} 1, & c_i = c_j \\ 0, & c_i \neq c_j \end{cases}
$$

where $\gamma$ is a resolution parameter ($\gamma = 1$ in the original formulation). $Q$ ranges from $-1$ to $1$; higher values indicate a stronger community structure, i.e. many more edges inside clusters than would be expected by chance. For each Leiden resolution $r \in \{0.1, \ldots, 1.0\}$ we build the 10-NN graph in 40-dimensional PCA space; run Leiden with that resolution; Convert the graph to igraph and call g.modularity(labels, weights=g.es["weight"]).

Advantages: 1. Native to graph-based clustering. Leiden and Louvain maximize modularity during optimization, so reporting $Q$ provides an internal goodness-of-fit measure for exactly the objective the algorithm tries to optimize. 2. Fast to compute. Once the cluster assignment is known, $Q$ is $O(|E|)$ and supported by efficient implementations in igraph and graph-tool. 3. Resolution tuning via $\gamma$. Varying $\gamma$ (or the resolution argument in Scanpy/Seurat) directly changes the trade-off between cluster granularity and modularity gain, allowing users to plot an elbow and pick the peak. 4. No feature-space assumptions. Unlike distance-based indices, modularity depends only on the graph and is therefore agnostic to the original dimensionality or scaling of the data.

**Functional enrichment analysis** For every cluster selected at the specific Leiden resolution we queried the Gene Ontology (GO) Biological-Process 2023 library. Using the Fisher-exact framework, we obtained a ranked list of GO terms with Benjamini–Hochberg–adjusted $P$-values ($P_{\text{adj}}$). Terms with $P_{\text{adj}} < 0.05$ were considered significant. We just take the top five functions sorted by $P_{\text{adj}}$.

# E    Supplementary Figures

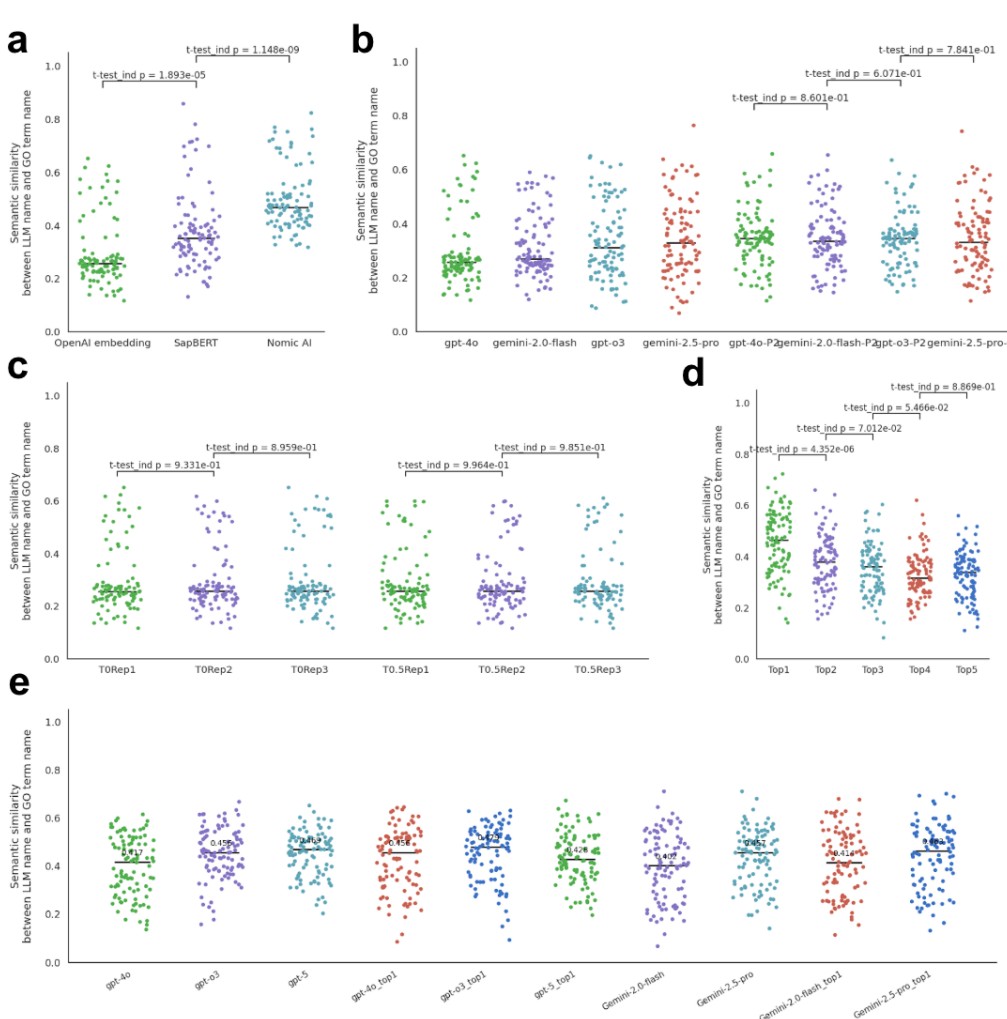

Figure S1: Benchmark for different parameters. (**a**) Embedding-method comparison. (**b**) General prompt V1 vs. V2. (**c**) Temperature sweep and repeats for GPT-4o. (**d**) Top-5 candidate analysis on GPT-o3. (**e**) General V2 vs. hypothesis prompt across models.

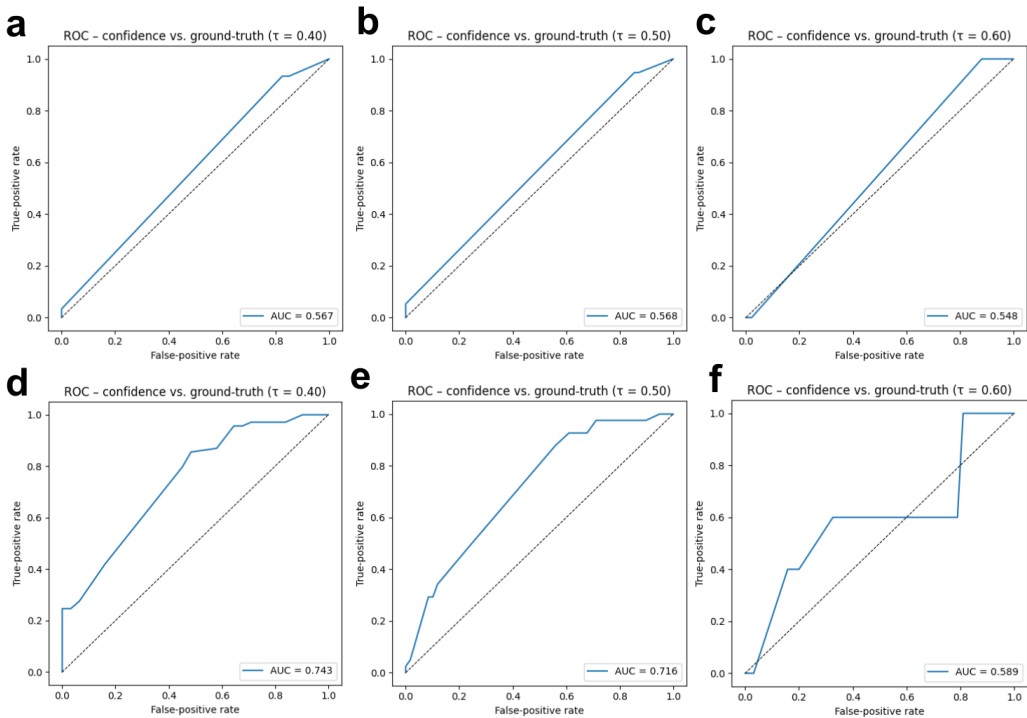

Figure S2: AUC metric for GPT-4o top 1 group performance (**a,b,c**) and GPT-o3 top 1 group performance (**d,e,f**) at different thresholds (0.40, 0.50, 0.60).

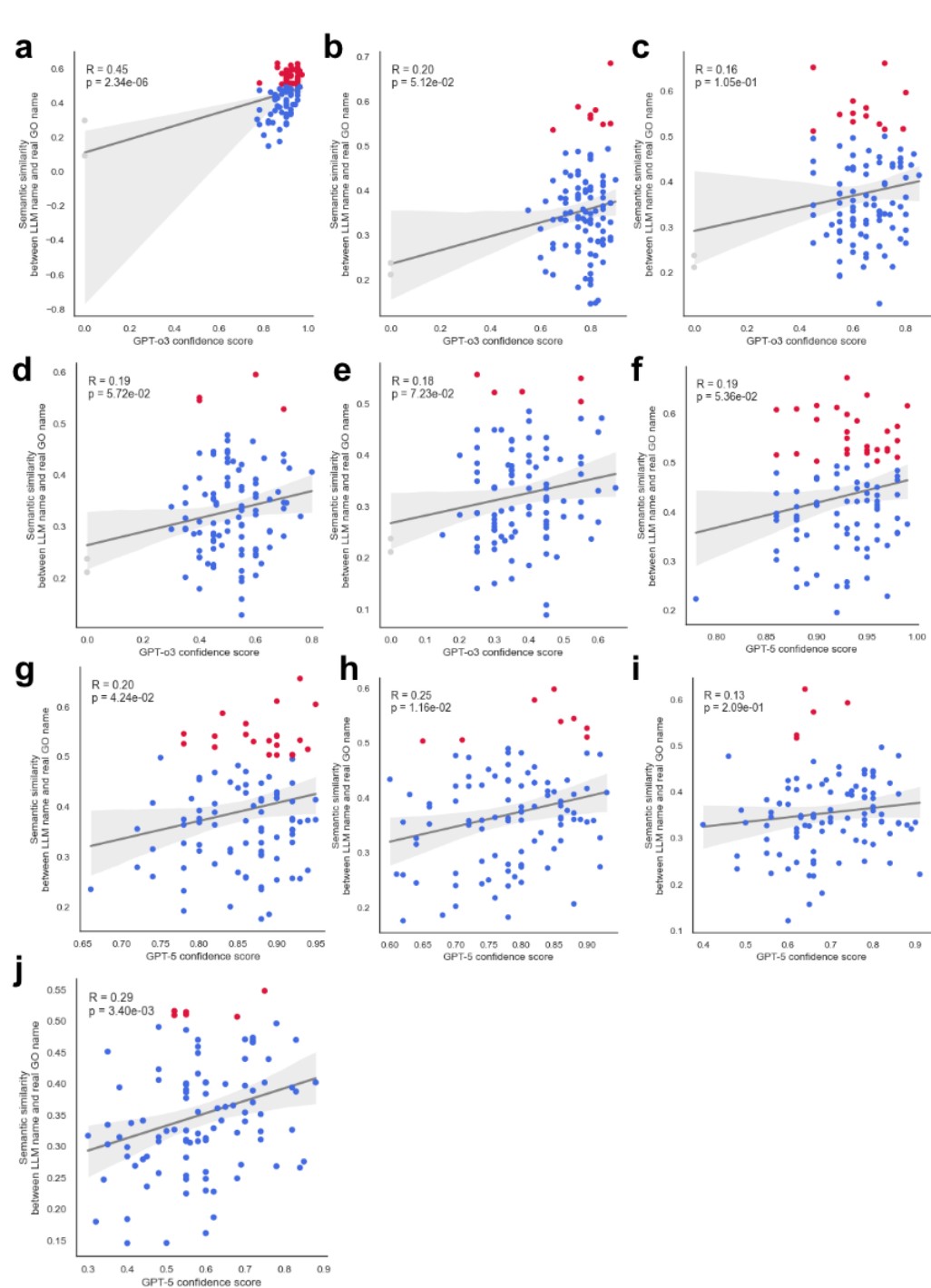

Figure S3: Consistency comparison between similarity score  model's confidence score. (**a**) Top1 (**b**) Top2 (**c**) Top3 (**d**) Top4 (**e**) Top5 group candidates of GPT-o3 model; (**f**) Top1 (**g**) Top2 (**h**) Top3 (**i**) Top4 (**j**) Top5 group candidates of GPT-5 model.

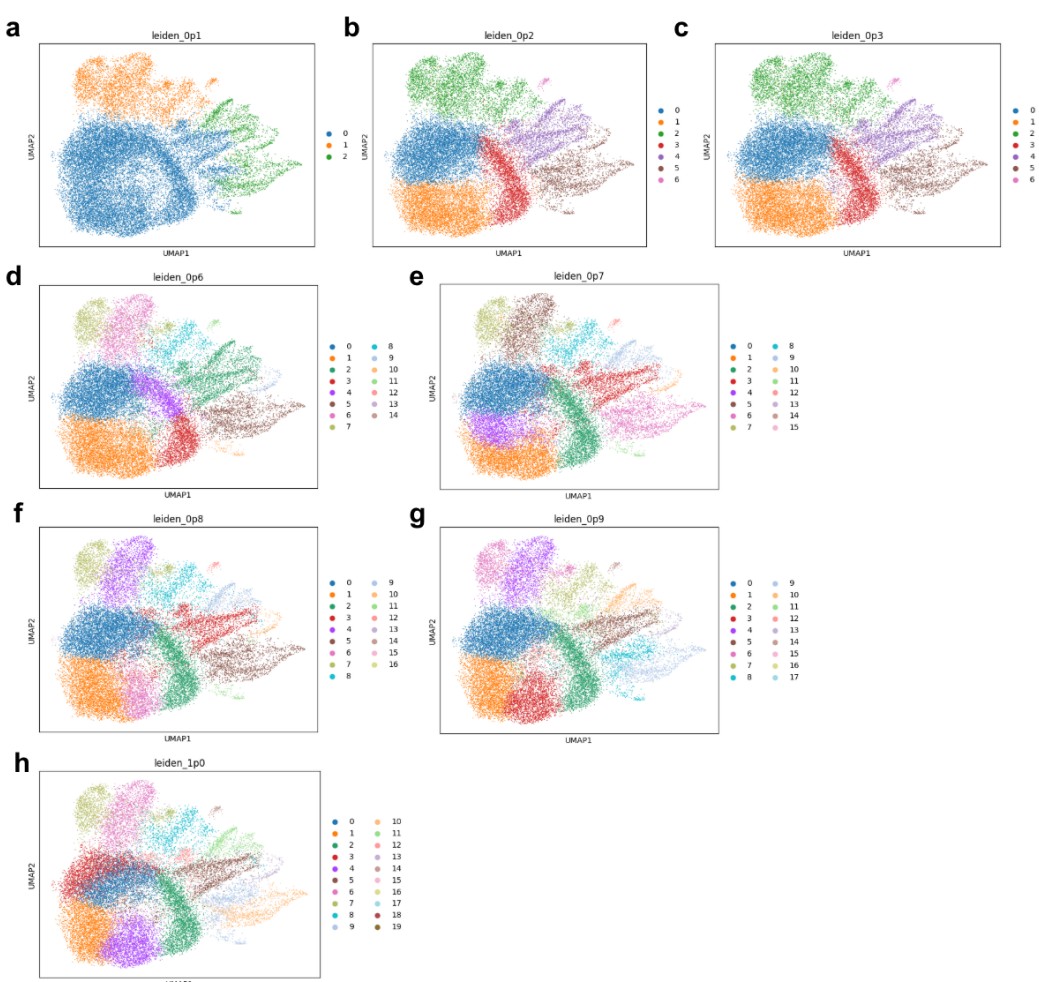

Figure S4: UMAP of the perturb-seq dataset at different resolutions. (**a**) r = 0.1, 3 clusters. (**b**) r = 0.2, 7 clusters. (**c**) r = 0.3, 7 clusters. (**d**) r = 0.6, 15 clusters. (**e**) r= 0.7, 16 clusters. (**f**) r = 0.8, 17 clusters. (**g**) r = 0.9, 18 clusters. (**h**) r = 1.0, 20 clusters.

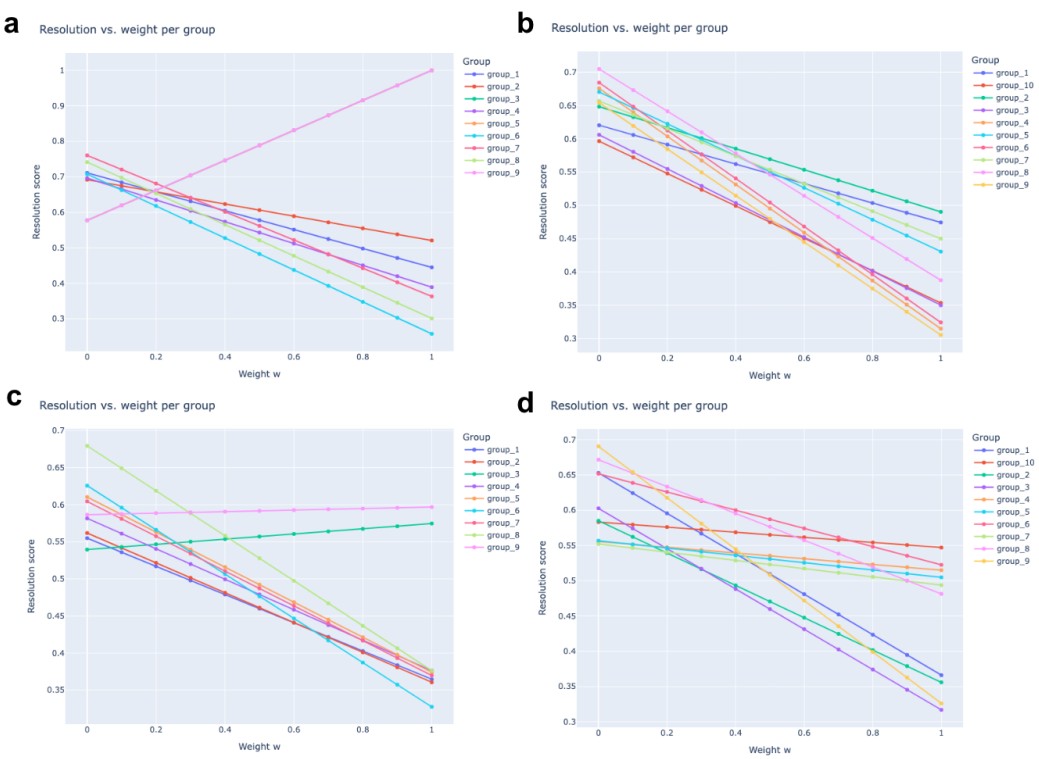

Figure S5: Hyper parameter w tests in [0,1]. (**a**) GEX cluster level, Leiden resolution is 0.4, group 9 (cluster 8) is the outlier (**b**) GEX cluster level, Leiden resolution is 0.5 (**c**) perturbation cluster level, Leiden resolution is 0.4, group 3 (cluster 2) and group 9 (cluster 8) are the outliers (**d**) perturbation cluster level, Leiden resolution is 0.5.

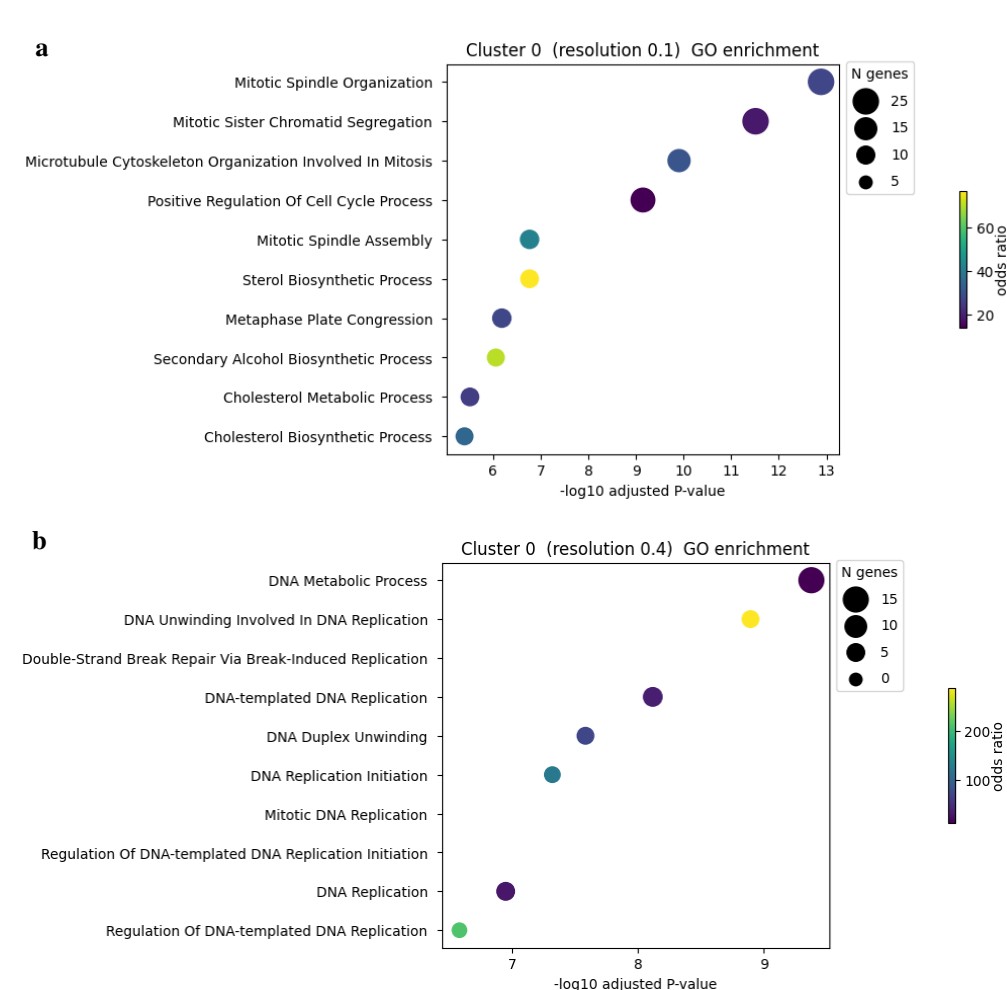

Figure S6: Gene Ontology (GO) enrichment analysis. **(a)** Dot plot of the gene enrichment result of cluster 0 at resolution 0.1. **(b)** Dot plot of the gene enrichment result of cluster 0 at resolution 0.4.

# F  PROMPT ENGINEERING

**General prompt V1 for GOBP test**

> system:
> Generates a critical analysis of the biological processes performed by a system of interacting proteins.
> Instructions:
> 1. Base the analysis on known biological roles and interactions of the proteins.
> 2. Identify and describe the most prominent biological process performed by the system.
> 3. Assign a confidence score between 0.00 and 1.00 based on how well the proteins support the named process.
> Guidelines:
> - Avoid vague or generic process names (e.g., "Cellular Signaling").
> - Avoid listing unrelated protein facts.
> - Focus on integration, synergy, and process coherence among the proteins.
> - If no prominent process is supported, return:
> Process: System of unrelated proteins (0.00)
> Output Format:
> - Title: Process: <name >(<score >)
> - Paragraphs: Factual, concise description of the protein interactions and biological function.
> Example:
> Input:
> proteins = ["PDX1", "SLC2A2", "NKX6-1", "GLP1", "GCG"]
> Output:
> Process: Pancreatic development and glucose homeostasis (0.96)
> 1. PDX1 is a homeodomain transcription factor involved in the specification of the early pancreatic epithelium and its differentiation. It activates genes like insulin and glucose transporter 2 (SLC2A2), supporting pancreatic beta-cell identity.
> 2. NKX6-1 regulates beta-cell development during secondary transition and works alongside other neural and pancreatic regulators.
> 3. GCG and GLP1 modulate glucose levels. GCG increases glucose via gluconeogenesis; GLP1 enhances insulin release and beta-cell mass.
> 4. SLC2A2, encoding GLUT2, supports glucose uptake in beta-cells and hepatocytes. Together, these proteins define the endocrine pancreas and maintain glucose homeostasis through synergistic regulation of transcription, metabolism, and hormone signaling.

**General prompt V2 for GOBP test**

system:
Generate an integrative, evidence-based analysis of a set of interacting genes.
INSTRUCTIONS
1. Use well-established knowledge from major public resources (e.g.PubMed, UniProt, GeneCards, KEGG) to extract the most relevant, non-trivial information for every gene in the list.
2. For EACH gene supply:
• 2-5 concise, factual bullets of its best-known molecular roles, pathways, or interaction partners.
• One plain-language, $\leq$ 20-word sentence that captures its essence ("One-line summary").
3. After all genes are described, propose an overarching biological process / pathway that logically unifies the set.
4. Rate your confidence in that process with a score between 0.00–1.00, where 1.00 = overwhelming support and 0.00 = no support.
5. If the genes do not clearly converge on any coherent process, output exactly:
Process: System of unrelated genes (0.00)
STYLE & QUALITY GUIDELINES
• Prefer specific pathway names (e.g. "Wnt/$\beta$-catenin–driven osteogenesis") over vague terms ("cell signalling").
• Highlight functional synergy and cross-talk among the genes; avoid isolated fact lists.
• Keep language factual, concise, and free of speculation.
• Use markdown; inline math with single $ . . . $, block math with double $$.
• Do NOT invent citations, but you may mention reputable databases in prose (e.g. "(KEGG)").
OUTPUT TEMPLATE
Title: Process: <Descriptive name >(<Confidence score >)
Genes
<Gene 1 >
- Fact bullet 1
- Fact bullet 2
- . . .
One-line summary: <simple sentence >
<Gene 2 >
. . .
One-line summary: . . .
(repeat for all genes)
Overall synthesis
<Short paragraph explaining how the genes work together in the named process. Emphasise mechanistic links, pathways, and complementary roles. >

**Hypothesis prompt for GOBP test (propose top5 candidates)**

system:
You are a biomedical assistant. Follow all OpenAI policy.
Generate an integrative, evidence-based analysis of a set of interacting genes.
INSTRUCTIONS
1. Use well-established knowledge from major public resources (e.g.PubMed, UniProt, GeneCards, KEGG) to extract the most relevant, non-trivial information for every gene in the list.
2. For EACH gene supply:
• 2-5 concise, factual bullets of its best-known molecular roles, pathways, or interaction partners.
• One plain-language, $\leq$ 20-word sentence that captures its essence ("One-line summary").
3. After all genes are described, propose up to **five** distinct biological process / pathway names that could plausibly unify the set.
4. Rank them from highest to lowest confidence and give each a numeric score between 0.00 and 1.00.
5. If the genes do not clearly converge on any coherent process, output exactly: Process: System of unrelated genes (0.00)
STYLE & QUALITY GUIDELINES
• Prefer specific pathway names (e.g. "Wnt/$\beta$-catenin–driven osteogenesis") over vague terms ("cell signalling").
• Highlight functional synergy and cross-talk among the genes; avoid isolated fact lists.
• Keep language factual, concise, and free of speculation.
• Use markdown; inline math with single $ ... $, block math with double $$.
• Do NOT invent citations, but you may mention reputable databases in prose (e.g. "(KEGG)").
OUTPUT TEMPLATE
Analysis
<Concise paragraph of evidence >
Top-5 candidate processes
1. <Process name 1 >— <Confidence 1 >
2. <Process name 2 >— <Confidence 2 >
3. . . .
5. <Process name 5 >— <Confidence 5 >

**Hypothesis prompt for Replogle Perturb-seq test (propose top5 candidates)**

system:
You are a biologist analyzing gene sets. Given a list of genes, describe what these genes do and generate up to 5 one-sentence descriptions that represent the biological processes these genes are most likely responsible for.

The gene sets are from different clusters of a single-cell RNA-seq dataset, and they are all related to the same experiment.

The dataset is from human K562 cell line of 53 year old female with chronic myeloid leukemia disease, treated with perturbation type "CRISPR".

INSTRUCTIONS

1. Use well-established knowledge from major public resources (e.g. PubMed, UniProt, GeneCards, KEGG) to extract the most relevant, non-trivial information for every gene in the list.

2. For EACH gene supply:
• 2-5 concise, factual bullets of its best-known molecular roles, pathways, or interaction partners.
• One plain-language, $\leq$ 20-word sentence that captures its essence ("One-line summary").

3. After all genes are described, propose up to **five** distinct biological process / pathway names that could plausibly unify the set.

4. Rank them from highest to lowest confidence and give each a numeric score between 0.00 and 1.00.

5. If the genes do not clearly converge on any coherent process, output exactly: Process: System of unrelated genes (0.00)

STYLE & QUALITY GUIDELINES

• Prefer specific pathway names (e.g. "Wnt/$\beta$-catenin–driven osteogenesis") over vague terms ("cell signalling").

• Highlight functional synergy and cross-talk among the genes; avoid isolated fact lists.

• Keep language factual, concise, and free of speculation.

• Use markdown; inline math with single $ . . . $, block math with double $$.

• Do NOT invent citations, but you may mention reputable databases in prose (e.g. "(KEGG)").

OUTPUT TEMPLATE

Analysis

<Concise paragraph of evidence >

Top-5 candidate processes

1. <Process name 1 >— <Confidence 1 >
2. <Process name 2 >— <Confidence 2 >
3. . . .
5. <Process name 5 >— <Confidence 5 >

