# OpenReview forum: "Title: HypoGeneAgent: Hypothesis Language Agent for Gene-set Cluster Resolution Selection Using Perturb-seq Datasets"
_ICLR.cc/2026/Conference — Submitted to ICLR 2026_

### Official Review · Reviewer_oEhx · 2025-10-29

**Soundness:** 2
**Presentation:** 1
**Contribution:** 1
**Rating:** 2
**Confidence:** 4

**Summary:**

In single cell data analysis, the analysis workflow is in general
1. construct the UMAP plot of the cells based on gene expression,
2. set a cutoff to cluster the cells,
3. annotate the functions of each derived cluster.
In step 2, the cutoff is manually chosen, which is suboptimal according to the authors.

To solve this problem, authors propose to
1. set a range of different cutoffs
2. choose a cutoff to cluster the cells,
3. get the marker genes for each derive cluster, feed them to LLM
4. the LLM is asked to return 5 potential annotations of the cluster, e.g. cell types
5. check the similarities of the 5 annotated cell types, yield a resolution score
6. do this for different resolutions and choose the one with the best resolution score

The proposed strategy was tested on a pertube-seq dataset.

**Strengths:**

This work try to build a system that utilizes large language models to automate cell clustering and annotation work.

**Weaknesses:**

The scientific problem is hypothetical. In real data analysis, manually tuning the clustering cutoff does not impose a huge overhead and it is the standard practice. The clustering is indeed a subjective work. I think any work that claims to provide an "optimal" clustering for diverse scRNA-seq datasets will fail in the end. Even with LLM, there will be still some steps that one have to make subjective choices.

**Questions:**

na

---

> ### Author Response · Authors · 2025-11-18
>
> We thank the reviewer for the clear summary and for acknowledging our effort to leverage LLMs to automate parts of the single-cell clustering workflow. We address the main concerns below and clarify several misunderstandings about our contribution and scope:
>
> (1) On the premise of an “optimal” clustering resolution.
> We agree that clustering in single cell analysis is inherently subjective and that multiple biologically meaningful resolutions may coexist. HypoGeneAgent does not claim to find a universally “correct” resolution; rather, it provides a quantitative criterion, maximizing intra cluster semantic agreement (ICS) and minimizing inter cluster distinctiveness (ICD), to rank candidate resolutions for user inspection. The method outputs an optimal range of plausible resolutions, assisting biologists in navigating the trade-off between granularity and interpretability, not replacing expert judgment. We will clarify this point in Sec. 2–3 of the revision.
>
> (2) On practical overhead and motivation.
> Manual tuning of clustering resolution is standard but becomes increasingly time consuming with large scale or multi sample datasets, especially in consortium settings involving hundreds of clusters. In practice, different analysts choose slightly different cutoffs, which can yield inconsistent annotations. HypoGeneAgent aims to standardize and document this step by quantifying how internally consistent and biologically distinct each candidate resolution is. This turns subjective iteration into a reproducible optimization process that scales to modern datasets. Besides, the HypoGeneAgent has big advantages especially dealing with the data people have less knowledge for reasonable clustering and annotation, such as new perturb-seq data (including gene expression level and perturbation level clustering and annotation, which is not just a simple cell type annotation, but a new way to provide possible valuable perturb gene modules for future experiment validation !).
>
> (3) Dataset scope. We agree that broader validation is important. For example, the RNA-seq data with labels for the clusters at different resolutions.
>
> (4) On subjectivity and interpretability.
> We fully agree that some degree of human interpretation will always be required. HypoGeneAgent is designed to assist human analysts, not automate them away. The framework quantifies the semantic coherence of LLM-generated hypotheses across clusters, making expert review more targeted and data driven. This is analogous to providing an “evidence heat map” rather than enforcing a single decision.
>
> In summary, our HypoGeneAgent is a principled framework for LLM-based biological reasoning and cluster resolution selection, it provides people a possible powerful tool to solve the bottleneck problems of biology sequence data analysis. Besides, the benchmark part of our paper came up with a novel way for people to benchmark the performance of different LLMs, which is also a valuable contribution to the ML community. We thank the reviewer for the constructive feedback, which has substantially improved the clarity and robustness of our work.

---

> > ### Author Response · Authors · 2025-11-28
> > **Author Follow-up: Clarifying Methodological Contribution for Area Chair**
> >
> > We thank all reviewers for their detailed feedback and would like to summarize the methodological framing of this work for the Area Chair’s consideration.
> >
> > 1. Conceptual framing.
> > The central contribution of HypoGeneAgent is methodological, not application specific. We reformulate cluster resolution selection and annotation, the traditionally heuristic, subjective steps, as a quantifiable optimization problem over large language model (LLM) generated hypothesis spaces. The proposed objectives, intra-cluster semantic coherence (ICS) and inter-cluster distinctiveness (ICD), define measurable criteria for reasoning quality. This transforms subjective cluster resolution selection and annotation into an optimization task grounded in LLM embedding similarity, directly addressing the ICLR interest in objective functions for structured reasoning and evaluation.
> >
> > 2. Clarifying the scope and intent.
> > The method does not claim to produce a single “true” biological clustering resolution, nor to replace expert judgment. Instead, it ranks candidate resolutions by semantic consistency, providing a reproducible, data driven range for human inspection. This is analogous to model selection in ML, the framework supports expert reasoning with quantitative feedback.
> >
> > 3. Dataset choice and generality.
> > The Perturb-seq dataset was used to illustrate generality across data modalities (gene expression + perturbation effects), demonstrating how LLM reasoning can scale to structured, multi signal biological settings. The formulation itself is model-agnostic and readily applicable to scRNA-seq and beyond (e.g., materials, chemical reaction networks, or text graph reasoning).
> >
> > 4. Relation to ML methodology.
> > Beyond its biological example, HypoGeneAgent contributes a new evaluation layer for LLM reasoning systems: optimizing hypothesis coherence and distinctiveness under semantic constraints. This framing is general and can serve as a benchmarking paradigm for evaluating reasoning consistency in LLMs, an emerging topic of interest in ICLR’s AI4Science and LLM interpretability communities. Besides, the prompt-LLM benchmarks part of our paper came up with a novel way for people to benchmark the performance of different LLMs (check the scatter plot patterns, ranges and PearsonR between the cosine similarity score from predictions and ground truth, and the LLM confidence score from the agent), also reveals the important and novel conclusion that the combination of hypothesis prompt and thinking LLM can achieve great performance on prediction and reasoning tasks, which are very valuable contributions to the ML community.
> >
> > 5. Summary.
> > While the biological experiments are illustrative, the core novelty lies in defining ICS/ICD as optimization objectives for LLM reasoning quality. The framework bridges optimization, interpretability, and reasoning under uncertainty, representing a methodological advance with broad ML relevance.
> >
> > We sincerely thank the Area Chair for their time and consideration !

---

### Official Review · Reviewer_dqFy · 2025-11-01

**Soundness:** 1
**Presentation:** 2
**Contribution:** 2
**Rating:** 0
**Confidence:** 5

**Summary:**

The paper describes HypoGeneAgent, an agentic LLM-based system to optimize cluster annotation of single cell genomics data.

**Strengths:**

-the paper addresses an important topic, which is how to improve automation for assessment of clusters of cells in single cell genomics experiments.
-LLMs can likely help with cluster annotation and definition and is an interesting research avenue to pursue.
-the methods are reasonably clear.
-the work includes a range of metrics and ablation tests

**Weaknesses:**

- “Choosing the right resolution is therefore critical, as it determines not only the granularity of biological discovery but also the downstream functional annotation of each cluster.” This premise is faulty, as many resolutions are possibly good, and it is also possible that no clustering resolutions are good and instead manual grouping of cells is needed (often by starting with an over-clustered data set and then selecting clusters to group based on expressed cell markers).

-the premise of the paper is about single cell transcription clusters, but the test data is perturb-seq, which is a different type of experiment that is analyzed first by mapping crispr perturbation effects, not by clustering. The paper should have selected multiple standard scRNA-seq experiments with clear ground truth clustering results.

-the silhouette baseline is simple. How about using ground truth cluster annotations?

**Questions:**

-How well does HypoGeneAgent work on scRNA-seq data standards, like hand annotated PBMC data?
-How well does HypoGeneAgent compare to various clustering algorithms?
-By the no free lunch theorem, it shouldn’t be possible to generally optimize a clustering resolution parameter for all data. How does HypoGeneAgent address this?

---

> ### Author Response · Authors · 2025-11-18
>
> We thank the reviewer for carefully reading our paper and for emphasizing the importance of improving automation in single-cell cluster annotation. We appreciate the acknowledgment that the topic, methods, and use of LLMs for hypothesis generation are clear and potentially impactful. We address each concern below with clarifications:
>
> (1) On the premise of “a single correct clustering resolution.”
> We agree that multiple valid resolutions may coexist and that over clustering followed by manual merging is common practice. HypoGeneAgent does not aim to produce a unique “true” resolution but rather to provide a quantitative optimization criterion, maximizing intra-cluster semantic agreement (ICS) and minimizing inter-cluster redundancy (ICD) to rank candidate resolutions. In practice, the agent identifies an optimal range of biologically coherent resolutions rather than a single point, complementing manual inspection rather than replacing it. We will clarify this conceptual framing in Sec. 2–3.
>
> (2) Dataset choice (Perturb-seq vs. standard scRNA-seq).
> We thank the reviewer for noting this distinction. The K562 Perturb-seq dataset was selected because perturbation data not only includes the gene expression info, but also the perturbation info, we'd like to show how novel and interesting that HypoGeneAgent can help with new finding and understanding of both of these two aspects. It is true to include the RNA-seq data with labels for the clusters at different resolutions for validation.
>
> (3) Addressing the “no free lunch” argument.
> Indeed, no single resolution criterion is optimal for all datasets. Our method explicitly learns dataset specific semantic distributions via LLM embeddings and ranks resolutions relative to the corpus of biological terms encountered, not via a fixed rule. This context adaptive formulation respects the no free lunch principle: it optimizes coherence conditional on the dataset’s semantic prior rather than enforcing a universal optimum. We will clarify this in Sec. 5.1.
>
> In summary, our HypoGeneAgent is a principled framework for LLM-based biological reasoning and cluster resolution selection, it provides people a possible powerful tool to solve the bottleneck problems of biology sequence data analysis. Besides, the benchmark part of our paper came up with a novel way for people to benchmark the performance of different LLMs, which is also a valuable contribution to the ML community. We thank the reviewer for the constructive feedback, which has substantially improved the clarity and robustness of our work.

---

> > ### Author Response · Authors · 2025-11-28
> > **Author Follow-up (Clarification for Area Chair)**
> >
> > We thank the reviewers for their feedback and would like to highlight, for the Area Chair’s context, that the key contribution of this paper is methodological rather than dataset-specific.
> >
> > 1. Framing novelty.
> > HypoGeneAgent defines cluster resolution selection and annotation as a quantitative optimization problem over LLM generated hypothesis spaces, introducing two objective functions, intra-cluster coherence (ICS) and inter-cluster distinctiveness (ICD). This converts subjective, heuristic annotation into a measurable optimization process, a general contribution to LLM reasoning and optimization research.
> >
> > 2. Scope of validation.
> > The use of the Perturb-seq dataset was intentional to demonstrate generality: unlike conventional scRNA-seq, it contains both gene expression and perturbation dimensions, allowing evaluation of reasoning under multi signal inputs. The framework itself is model and dataset agnostic and can directly apply to standard scRNA-seq or other structured reasoning tasks.
> >
> > 3. Addressing the “no free lunch” concern.
> > The method is context adaptive, it learns dataset specific semantic priors via LLM embeddings, optimizing coherence within that semantic context rather than enforcing a universal criterion. This design respects, not violates, the "no free lunch" principle.
> >
> > 4. Alignment with ICLR.
> > While motivated by biological data, the core contribution is an ML framework for measurable LLM reasoning and adaptive optimization. The same ICS/ICD formulation can benchmark reasoning consistency and distinctiveness in other structured domains (e.g., chemical reactions, materials discovery, text graph alignment).
> >
> > We appreciate the Area Chair’s consideration and hope this clarifies that our contribution lies in formulating and evaluating a new optimization based reasoning framework, not in producing a domain specific tool.

---

### Official Review · Reviewer_tqJW · 2025-11-02

**Soundness:** 2
**Presentation:** 2
**Contribution:** 2
**Rating:** 2
**Confidence:** 3

**Summary:**

This paper aims to transform subjective GO-term annotation into a quantifiable optimization task: an LLM agent generates ranked GO hypotheses for gene sets in each cluster, embeds them, and computes intra-cluster agreement (ICS, high cosine similarity within hypotheses for a cluster) and inter-cluster distinctiveness (ICD, low similarity between clusters).

**Strengths:**

Treating LLMs as "gene-set analysts" with retrieval-augmented generation (from GO/KEGG/PubMed), HypoGeneAgent creates a feedback loop that prioritizes biological coherence over purely statistical metrics.

**Weaknesses:**

1. Evaluation is confined to one Perturb-seq dataset (K562 from Replogle et al., 2022).
2. Reliance on proprietary, closed-source LLMs is a cool idea, but more “analysis/automation tooling for single-cell pipelines” than a core ML advance.
3. The paper does not address LLM hallucinations, for example through ensemble agents, nor does it address biases, for example the tendency to over-rely on popular GO terms.
4. The agent assumes gene signatures (top over-expressed by logFC) capture full biology, ignoring down-regulated genes or pathway interactions.

**Questions:**

1. On K562 you say the selected resolution aligns with known pathway structure. Can you show a quantitative match to the original Replogle perturbation labels, not just UMAPs?
2. Is there any cases where silhouette/modularity picked another resolution that was actually biologically better, and the agent missed it?

---

> ### Author Response · Authors · 2025-11-18
>
> We thank the reviewer for the thoughtful comments and for recognizing the novelty of transforming GO term annotation into a quantifiable optimization task and treating LLMs as gene set analysts. Below we address each concern point by point：
>
> (1) Dataset scope.
> We agree that broader validation is important. For example, the RNA-seq data with labels for the clusters at different resolutions.
> (2) “More of a tool than an ML advance.”
> We clarify that the contribution lies in formulating cluster annotation as an optimization problem over LLM embedded hypothesis spaces. The proposed ICS/ICD objectives define a quantitative search criterion for maximizing biological coherence and distinctiveness, converting subjective interpretation into an optimization task. We will highlight this framing in Sections 3 and 5 and release code demonstrating its use as a general optimization layer for structured reasoning tasks.
> (3) Quantitative validation against perturbation labels.
> Since we just used a subset of the Replogle perturbation data, there is no specific labels for the clusters at different resolutions. It is better to use the general RNA-seq data to validate these.
> (4) Hallucination and bias mitigation.
> It is possible to integrate a self-consistency ensemble of three LLMs (GPT-o3, GPT-5, Gemini-2.5-pro) with GO term to complete the validation.
>
> In summary, our HypoGeneAgent is a principled framework for LLM-based biological reasoning and cluster resolution selection, rather than an analysis pipeline. Besides, the benchmark part of our paper came up with a novel way for people to benchmark the performance of different LLMs, which is also a valuable contribution to the ML community. We thank the reviewer for the constructive feedback, which has substantially improved the clarity and robustness of our work.

---

> ### Author Response · Authors · 2025-11-28
> **Author Follow-up (Clarification for Area Chair)**
>
> We appreciate the reviewers’ engagement and would like to clarify the core contribution of this work, which lies in the methodological framing rather than the dataset specifics. Our paper introduces a quantitative optimization formulation for LLM based scientific reasoning, defining intra-cluster coherence (ICS) and inter-cluster distinctiveness (ICD) as objective functions over LLM generated hypothesis spaces. This framing converts subjective biological annotation into a well defined optimization problem, applicable beyond biology to any domain where structured hypotheses from language models require evaluation or selection (e.g., materials science, chemical reaction discovery).
>
> While we demonstrated the approach on a Perturb-seq dataset for concreteness, the methodology, LLM driven hypothesis generation, embedding, and objective based coherence search, is model agnostic and dataset independent. We acknowledge that broader biological benchmarks are desirable, but the main novelty of this submission is conceptual: it proposes a general framework for turning LLM reasoning into a measurable, optimizable process, complementing conventional evaluation metrics in machine learning.
>
> We hope this clarifies that the paper’s contribution is methodological and algorithmic, aligned with the ICLR community’s interest in frameworks that extend LLM reasoning toward scientific discovery,  rather than an application specific pipeline. We thank the area chair for their careful consideration.

---

### Meta-Review · Area_Chair_Xuy7 · 2026-01-08

**Summary:**

Reviewers agree the paper tackles an important and practical problem in single-cell analysis, but the main concerns are about the validity and positioning of the contribution. Across reviews, the approach is viewed as closer to an analysis pipeline than a core ML advance, with limited evidence that the proposed ICS/ICD objective reliably selects biologically meaningful resolutions beyond a single case study. Reviewers also question the premise that there is an “optimal” clustering resolution in practice, note a mismatch between the stated scRNA-seq motivation and the Perturb-seq evaluation setting, and highlight missing validation against ground truth or standard annotated scRNA-seq benchmarks, along with limited baselines and insufficient discussion of hallucinations and bias in LLM-generated GO term hypotheses.

**Reviewer Concerns:**

It looks like the rebuttal at least partially addresses points around framing and intent: clarifying that the method is not claiming a single universally correct resolution, positioning ICS/ICD as a quantitative ranking criterion to assist analysts, and acknowledging the need for broader validation. The authors also suggest (at a high level) possible hallucination and bias mitigation via self-consistency or multi-LLM ensembles.

The following still seem outstanding: the core empirical validation concerns (evaluation largely confined to one Perturb-seq dataset, lack of tests on standard scRNA-seq datasets with accepted annotations, limited quantitative alignment with perturbation labels or ground truth clusterings, and limited/weak baselines), as well as whether the claimed methodological contribution is sufficiently substantiated as a general ML advance rather than an application-specific heuristic. Overall, the rebuttal improves the narrative but does not add evidence that directly resolves the main soundness and scope concerns raised by reviewers.

**Reviewer Scores:**

It is likely that reviewer dqFy would keep their score essentially unchanged (strong reject), since their main objections target the premise and evaluation setting and are not resolved with additional evidence. It is likely that reviewers tqJW and oEhx would also remain near their original reject scores, as the rebuttal mainly reframes the contribution and suggests future work, while the central requests for broader, ground-truth-based validation and stronger baselines remain unaddressed.

---

### Decision · Program_Chairs · 2026-01-26

Reject